# Applications of Reflectance Confocal Microscopy in the Diagnosis of Fungal Infections: A Systematic Review

**DOI:** 10.3390/jof9010039

**Published:** 2022-12-26

**Authors:** Samavia Khan, Banu Farabi, Cristian Navarrete-Dechent, Babar Rao, Bijan Safai

**Affiliations:** 1Center for Dermatology, Rutgers Robert Wood Johnson Medical School, New Brunswick, NJ 08901, USA; 2Rao Dermatology, Atlantic Highlands, NJ 07095, USA; 3Dermatology Department, New York Medical College/Metropolitan Hospital Center, New York, NY 10029, USA; 4Department of Dermatology, Escuela de Medicina, Pontificia Universidad Católica de Chile, Santiago 8331150, Región Metropolitana, Chile; 5Department of Dermatology, Weill Cornell Medicine, New York, NY 10021, USA

**Keywords:** reflectance confocal microscopy (RCM), fungal infections, fungi, medical dermatology, non-invasive skin imaging

## Abstract

Cutaneous and adnexal fungal infections are typically diagnosed with potassium hydroxide (KOH) skin scrapings, fungal cultures, and Periodic acid-Schiff (PAS) biopsy staining. All three current methods of fungal diagnosis require sample processing and turnover time which leads to a delay in diagnosis. Reflectance confocal microscopy (RCM) is a non-invasive, in vivo skin imaging technology that provides real-time dermatologic diagnoses. We present an updated systematic review of the applications of RCM in diagnosing fungal infections in an effort to explore the utility of RCM as an adjunct clinical tool in detecting cutaneous and adnexal fungi We systematically searched the MEDLINE (via PubMed) for studies published from January 2000 to October 2022 that described the utility of RCM in the setting of fungal infections. Of the 25 studies that met the inclusion criteria, 202 patients were included. The following information on the application of RCM in the setting of fungal infections was extracted from each study, if reported: study type, year published, number of patients included, diagnosis/diagnostic methods, and RCM description. Concordant within all included studies, fungal infections presented on RCM as bright, linear, branching, filamentous structures at the level of stratum corneum. A limitation of this review is that 11 of 25 studies were case reports (n = 1). Larger scale studies should be conducted to explore the utility of RCM in diagnosing fungal infections and to enrich the RCM descriptions of specific fungal conditions.

## 1. Introduction

Cutaneous and adnexal fungal infections are a prevailing dermatologic concern worldwide and in the United States. Fungal infections are diagnosed with KOH skin scrapings, fungal cultures, and PAS biopsy staining. KOH scraping yields limited information about the viability of the isolated fungal spores and hyphae. If antifungal therapy has begun before KOH scraping, false negatives can be reported [1]. In addition, sampling error, excess scale in sample, and inadequate KOH hydrolysis may lead to false negative results. KOH scraping may also lead to false positive results as it may be difficult to distinguish fungal elements from contaminants (i.e., hair, cotton/wool fibers). Fungal culture can give insight into the viability of fungus and can distinguish fungal species; however, fungi are often fastidious and slow growing; thus, it can take days to weeks to obtain definitive results. Biopsies are invasive and can lead to discomfort and a disfiguring scar. In addition, there may be sampling errors depending on the location of the biopsy. All three traditional fungal diagnostic methods require a sample processing and turnover time and do not provide real-time results. A 2010 pooled analysis that used clinical assessment as the gold standard for diagnosis of tinea pedis, the sensitivities of KOH smear and culture were 73.3% (95% CI: 66.3 to 79.5%) and 41.7% (34.6 to 49.1%), respectively [2]. The specificities of culture and KOH smear were 77.7% (72.2 to 82.5%) and 42.5% (36.6 to 48.6%), respectively [2]. Due the low sensitivity and specificity of current diagnostic methods in fungal infections, it is worth exploring an adjunctive technology that can potentially streamline the diagnostic process.

Reflectance confocal microscopy (RCM) is a non-invasive imaging tool that visualizes skin and its appendages at near-histologic resolution (lateral resolution of 0.5 to 1 μm) [1]. RCM uses a low-power 830 nm diode laser to illuminate the targeted area of tissue. RCM produces grayscale horizontal images that can be taken at levels ranging from the stratum corneum to the superficial papillary dermis [1], corresponding to a depth of approximately 150–200 µm. An automated program allows the capture of a series of mosaics, which consist of single images 0.5 × 0.5 mm square, captured in a serpentine motion and then “stitched” together like a quilt to create a single, full-resolution image that is up to 8 × 8 mm square in a horizontal plane. Image contrast is provided by differences in the size and refractive indices of cellular organelles and extracellular structures. There are three commercially available RCM devices, Vivascope 1500, a wide-probe microscope, Vivascope 2500, an ex vivo imaging scope, and the Vivascope 3000, a handheld probe (Caliber Imaging and Diagnostics, Rochester, NY, USA). Prior to confocal microscopy imaging, a digital dermatoscopic image is obtained using the Vivacam^®^, a dermatoscopic camera that is integrated into the confocal platform. The program captures a dermatoscopic quality photo of the lesion that is also mechanically aligned to the confocal image and may be used to navigate the lesion.

RCM has shown utility in diagnosing various inflammatory conditions, melanocytic and non-melanocytic lesions. In addition, various studies have reported RCM’s utility in diagnosing fungal infections, as fungi take residence in the stratum corneum of the epidermis, an imageable layer by RCM. We present a systematic review of the literature on the applications of reflectance confocal microscopy (RCM) in diagnosing fungal infections and we assess the utility of RCM in diagnosing cutaneous and adnexal fungal infections in clinical dermatology. 

## 2. Materials and Methods

This systematic review was conducted in accordance with the Preferred Reporting Items for Systematic Reviews and Meta-Analyses (PRISMA) guidelines. 

### 2.1. Eligibility Criteria

We included all studies of any design which reported on applications of RCM in the setting of fungal infections. We included original peer-reviewed articles, literature reviews, single case-reports, conferences abstracts, and book chapters. 

### 2.2. Information Sources, Search, and Study Selection 

Screening, inclusion, and exclusion of studies for systematic review are outlined in Figure 1. Systematic literature searches were conducted (1 October 2022) in MEDLINE (via PubMed) database with no specified date, age, sex, or language restrictions. A bibliographic management tool (EndNote, Clarivate Analytics) was used to combine the search results and eliminate duplicates electronically. Search strategy employed the Medical Subject Headings (MeSH) phrases: ((“Microscopy, Confocal”[Mesh]) AND “Tinea”[Mesh]) AND “Dermatomycoses”[Mesh]) AND “Skin Diseases, Infectious”[Mesh]). References within the retrieved articles were also reviewed to identify other studies to include in the analysis. Two independent reviewers (BF and SK) screened the abstracts of identified studies and reviewed the full texts of those which were potentially eligible, with disagreements resolved by consensus. 

### 2.3. Data Collection and Extraction Process

The following information on the application of RCM in the setting of fungal infections was extracted from each study, if reported: study type, year published, number of patients included, diagnosis/diagnostic methods, and RCM description. All extracted data was recorded in an Excel spreadsheet (Microsoft Word). Across all studies, clinical/microbiological diagnosis and their corresponding RCM descriptions were compiled by the two independent reviewers (BF and SK). Meta-analysis was not performed due to the heterogeneity of the studies.

### 2.4. Quality Assessment

Two reviewers (BF and SK) independently performed a quality assessment of all studies. Reviewers used the Quality Assessment of Diagnostic Accuracy Studies-2 (QUADAS-2) tool to assess the risk of bias in the cohort and comparative studies [4]. Four risk of bias domains, including patient selection, the index test, the reference standard, and the flow of patients throughout the study, as well as three applicability concerns, including patient selection, index selection, and reference standard, were recorded. Risk of bias was then scored as ‘high,’ ‘low,’ or ‘unclear.’ Trials with high risk of bias were not to be included in the systematic review. 

## 3. Results

A total of 25 articles met the inclusion criteria and were included in the current study. Out of 25 studies, RCM was used in the diagnosis of fungal infections in 202 patients (excluding patients from 5 reviews). 

Table 1 details the QUADAS-2 risk for bias assessment of cohort and comparative studies included in this review. All such studies were assessed across four risk of bias domains and three applicability domains. Studies were generally determined as ‘low risk’ of bias with low concern regarding applicability of the evidence. Table 2 summarizes the studies, fungal diagnoses and corresponding RCM descriptions. Table 3 compiles the clinical/microbiological diagnosis with the corresponding RCM descriptions from across all 25 studies. 

Due to the presence of melanin in the fungal cell wall, the structures appear hyperreflective under RCM [18]. Fungi hyphae can be visualized as bright (hyper-refractive), linear, branching, and filamentous structures in the stratum corneum brighter than the background keratinized cells [9,20]. Conidia can be visualized as “hyper-refractive, small roundish bodies” [20,26]. Hyphae can mimic keratinocyte cell membranes and hair shafts and therefore, careful RCM evaluation is important [20]. On RCM of tinea manus, pedis, and cruris, inflammatory infiltrate and acanthosis was also visualized and described as “higher refractive polymorphonuclear cell populations, accompanied by light flashes, located in the stratum corneum or acanthosis” [5].

### 3.1. Visualization and Descriptions of Common Fungal Infections by RCM

RCM has been used to describe tinea incognito, dermatophytids, tinea capitis, tinea nigra, candida, tinea versicolor, and tinea incognito. Tinea incognito is a fungal infection that has been altered by the inappropriate use of topical or systemic corticosteroids. In a case report by Navarrete-Dechent et al. and a five-case series by Turan et. of tinea incognito, RCM was able to identify refractive linear septate branching structures, consistent with hyphae, in all cases [11,12]. Turan et al. noted that hyphae were seen at 445 μm to and the best images of fungal elements were obtained at 300 to 445 μm [10].

Additionally, RCM features of id reaction have been reported. An id reaction or dermatophytid is an eruptive allergic reaction to superficial fungal infection, including tinea cruris, tinea corporis, or tinea pedis [22]. Dermatophytid may present with pruritic a macular, papular and/or vesicular dermatitis. On RCM, dermatophytes were visualized in dermatophytid as “several bright linear structures (mycelium)” as well as inflammatory cells in the stratum corneum and in perivascular zones, as well as spongiosis.

RCM visualization of fungi in tinea capitis was shown to improve diagnosis and therapy. Tinea capitis may present as ectothrix (with Microsporum hyphae parasitizing on the outside of hair) or endothrix (with Trichophyton hyphae inside the hair shaft) [19]. Microsporum is treated with griseofulvin while trichophyton is treated with terbinafine. Veasey et al. used RCM to differentiate between microsporum and trichophyton in a diagnosis of tinea capitis in a child [19]. On RCM, hyperreflectivity of speckled appearing structures as well as tortuous structures were visualized outside of hair, consistent with Microsporum [19]. The patient was treated accordingly. It is important to note that dermatophytes have not been visualized within the medulla of the hair shaft [20].

Tinea nigra was visualized on RCM as linear and tortuous, irregular, and short speckled “hyphae” structures that were unlike the thin and elongated structures of dermatophytes [25]. Direct mycological examination confirmed tinea nigra with the *Hortae werneckii* species. RCM therefore allowed for differentiation of tinea nigra from other non-melanocytic lesions [25]. Mucormyetes were described as “hyper-reflective elongated 20 μm in diameter structures with perpendicular ramifications” in two cases of histology confirmed mucormycosis [17]. RCM was able to identify perifollicular inflammation in the diagnosis of majocchi’s granuloma, a dermatophytic infection of the dermal and subcutaneous tissue [23]. However, hyphae were not identified given the limitation of RCM’s penetration depth.

On RCM, candida is visualized with pseudo filaments and conidia in the skin, nails, and oral mucosa [20]. Malassezia furfur maintains a “spaghetti and meatball” morphology, similar to that of KOH scraping, with clusters of bright structures (“meatballs”) with torturous hyper-refractive structures, consistent with thick and short septa (“spaghetti”) [20].

In addition to cutaneous fungi, RCM is used to assess dystrophic nails in real time, at the bedside [18,19,20,21,22]. Antifungal treatments for onychomycosis have a long course and have serious, systemic side effects and therefore, it is important to confirm onychomycosis before treatment is initiated. Existing RCM scanner heads are not intended for nail examination [5]. However, Pharaon et al. used both Vivascope 3000 (handheld) and Vivascope 1500 (wide-pobe scope) to visualize onychomycosis in a prospective cohort study of 58 patients. Vivascope 3000 had a higher sensitivity (60%) than the Vivascope 1500 (50%), although the difference in sensitivity was not statistically significant. In addition, Vivascope 3000 was reported as requiring a similar length of time to complete nail examination to that required in completing KOH preparation. Pharaon et al. confirmed onychomycosis diagnosis on RCM based on the “presence of at least three consecutive images of bright filamentous branching structures corresponding to septate hyphae.” In addition, the presence of roundish structures in the nail plate corresponds to arthroconidia [6]. RCM was also used for non-invasive treatment monitoring for nine patients who received a full course of terbinafine [6]. At the end of treatment, all nine patients showed disappearance of bright filamentous structures and a restitution of the normal nail plate [6]. RCM additionally has shown similar hyper-reflective structures corresponding to hyphae and roundish homogeneous (5–10 μm in size) hyper-reflective structures corresponding to conidia in cases of tinea capitis and tinea barbae [26].

### 3.2. Ex-Vivo RCM

Leclercq et al. discuss the utility of ex vivo confocal microscopy, with the Vivascope 2500^®^, in diagnosing invasive fungal infections [17]. Vivascope 2500 ex vivo confocal microscopy (EVCM) allows real-time, quasi-histologic, microscopic examination of excised cutaneous tissue within a few minutes without frozen sectioning or traditional tissue processing, embedding, and sectioning. The device uses two diode lasers to visualize excised tissue biopsies in three different individualized modes: reflectance, fluorescence, and pseudocolor (digital hematoxyllin-eosin). Reflectance mode uses a 785nm wavelength to visualize subcellular structures by their differences in refractive indices. Fluorescence mode uses a 488nm wavelength to activate cellular fluorochromes to see cellular microstructures. Leclercq et al. observed mucormycetes as “hyper-reflective elongated 20μm in diameter structures with perpendicular ramifications” in two cases of histology-confirmed mucormycosis [17]. Leclercq noted that confocal microscopy was fast like the optical examination of skin scraping. In addition, no frozen sectioning or traditional tissue processing, embedding, and sectioning were required, therefore, there is a reduction in sample contaminants and false positives. The entire specimen of up to 2 cm in diameter could be observed on ex vivo confocal microscopy without further preparation beyond acridine orange staining, which lasts 2 to 3 min. No additional specimen segmentation is required, which allows for further histological examination as the specimen is not altered. Lastly, ex vivo RCM, similar to histology, allows for fungi localization within the tissue sample, unlike skin scraping and culture.

## 4. Discussion

This systematic review of 25 studies shows that consistent terminology, “bright (hyper-refractive), linear, branching, and filamentous structures in the stratum corneum,” are used to characterize fungi on RCM. RCM offers many advantages in diagnosing fungal infections. RCM offers a real-time, painless, non-invasive diagnosis that requires no additional sampling or turnover time. Although the time required for examination is similar to dermoscopy, RCM goes beyond dermoscopy in providing quasi-histological level details [18]. In the diagnosis of dermatophytid, RCM demonstrates its utility in rapidly examining multiple body locations non-invasively and in repeat imaging for therapeutic monitoring [7]. In addition, RCM is able to diagnosis inflammation and other quasi-histologic changes such as acanthosis, that are not visible by KOH scraping, akin to PAS stain on skin biopsies [5]. In a prospective study of 45 patients, Hui et al. showed that RCM was positive in 63.64% and 82.61% with the lesions of tinea manus and pedis and tinea cruris respectively, when microscopy was considered gold standard, which demonstrates good consistency between RCM and microscopic examination [5].

Limitations of RCM in diagnosing fungal infections include that the RCM device is expensive and thereby, may be less accessible for dermatologists. In addition, although the US Centers for Medicare and Medicaid Services granted reimbursement for RCM imaging of skin in 2017, many countries do not have reimbursement for imaging. Imaging requires an experienced imaging technician, particularly for difficult to image regions such as dystrophic nails and facial regions. Most of these areas require expertise using handheld RCM, which has a steeper learning curve than the tissue-coupled RCM. Although the RCM terminology used to define fungal functions are largely consistent across studies, an experienced image reviewer is still required to diagnose the presence of fungi.

In a review of RCM’s utility in detecting onychomycosis, RCM demonstrated a sensitivity of 52.9 to 91.67% and a specificity of 57.58 to 90.2% for detecting onychomycosis [6,21]. Pharaon et al. report a high specificity of 90.2% and attribute this to “the high spatial resolution of RCM (1.25 μm laterally and 5 μm vertically) that is smaller than the diameter of fungal hyphae (3–10 μm). Given this high sensitivity, Pharaon et al. recommend the initiation of topical or systemic antifungals if RCM is positive for fungi. However, the authors also report a low sensitivity (52.9%), for which they recommend that RCM should be used as an adjunctive tool and not on its own. Pharaon et al. discuss several limitations of RCM in diagnosing fungal infections. Firstly, RCM’s imaging depth is limited to 200μm. The average nail plate is around 500 μm and in the setting of onychomycosis, additional thickening of the nail plate may be observed. Therefore, RCM is limited in its ability to image thick, dystrophic nails. Secondly, because RCM is not intended for examination of the convex surface of the nail, it is difficult to obtain images that are well-positioned and show little movement. Pharaon et al. recommend technical improvements to improve the utility of the scope in performing nail examinations.

It is important to note that RCM is unable to differentiate fungal species and therefore, culture remains required when species identification is needed to guide therapy [16]. However, as demonstrated by Veasey et al. in a case of tinea capitis under RCM, RCM does illuminate quasi-histologic features of species that may help in narrowing the differential diagnosis. 11 of 25 studies included in this review were case reports (n = 1) and to truly enrich the RCM descriptions of specific fungal infections, larger scale studies should be conducted. Although RCM is far from serving as a single diagnostic modality or replacing current mycologic diagnostic methods, it shows great potential in elucidating the differential diagnosis and guiding further testing and therapeutic management.

## 5. Conclusions

Reflectance confocal microscopy allows for non-invasive visualization of fungal hyphae without additional sample turnover time. This review shows that tinea infections present on RCM as bright, linear, branching, filamentous structures at the level of stratum corneum. Further studies on RCM’s utility in diagnosing fungal infections should include larger study populations, refinement of RCM descriptive terminology for fungal infections, and technical advancements in the device to allow for nail examination in the setting of suspected onychomycosis.

## Figures and Tables

**Figure 1 jof-09-00039-f001:**
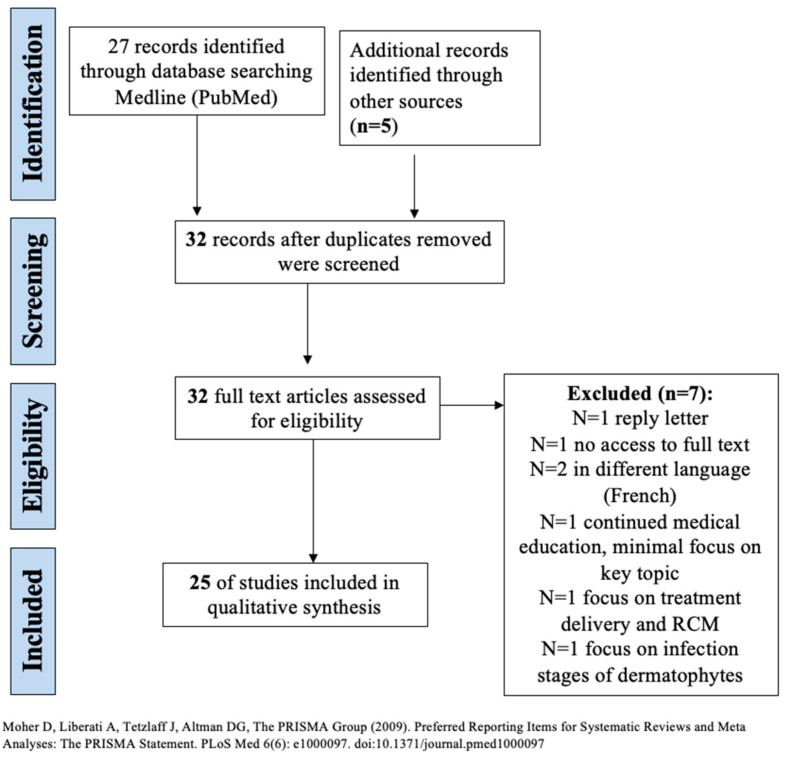
Screening, eligibility, inclusion, and exclusion of studies for systematic review [3].

**Table 1 jof-09-00039-t001:** QUADAS-2 Risk of Bias Assessment.

Study	Risk of Bias				Applicability Concerns		
	Patient Selection	Index Test	Reference Standard	Flow and Timing	Patient Selection	Index Test	Reference Standard
**Hui et al. [5]**	Low	Low	Unclear	Low	Low	Low	Unclear
**Pharaon et al. [6]**	Low	Low	Unclear	Low	Low	Low	Unclear
**Rothmund et al. [7]**	Low	Low	Low	Low	Low	Low	Low

**Table 2 jof-09-00039-t002:** **RCM of Fungal Infections.** Study characteristics and RCM descriptions summarized (N = 25).

Study	Year Published	Study Type	# of Patients	Diagnosis	Fungal Description (Adapted)
Slutsky et al. [8]	2011	Review	N/A	Tinea	Branching fungal hyphae
Hui et al. [5]	2013	Prospective cohort study	45	Tinea manus, pedis, cruris	Hyphae: bright linear branching structures not found in uninvolved skin Mycelium: a variable number of bright linear structures distributed in the stratum corneum Blister: round-low refractive of the structure located in the stratum corneum or acanthosis and inflammatory infiltrate: higher refractive polymorphonuclear cell populations, accompanied by light flashes, located in the stratum corneum or the upper layer of acanthosis
Liansheng et al. [9]	2013	Review	N/A	Tinea corporis	Highly refractile, septate, linear structures
Turan et al. [10]	2013	Case series	5	Tinea incognito	Hyphae: bright, thin, linear, and branched structures at stratum corneum Aggregation of bright, small structures resembling inflammatory cells
Cinotti et al. [11]	2014	Case Report	1	Tinea corporis	Bright, thin, linear, and branched structures
Cinotti et al. [12]	2014	Review	N/A	Onychomycosis	Lengthy structures with high reflection and the typical shape of hyphae
Mateus et al. [13]	2014	Case Report	1	Tinea pedis	Hyphae: Numerous linear bright structures
Pharaon et al. [6]	2014	Prospective Cohort Study	58	Onychomycosis	Presence of filamentous and/or roundish structures in the nail plate, corresponding respectively to septate hyphae and/or arthroconidia
Friedman et al. [14]	2015	Case Series	5	Tinea	Branching hyphae in the epidermis
Hoogedoorn et al. [15]	2015	Review	N/A	Tinea incognito, tinea corposis, onychomycosis	Branched hyphae network
Navarete-Dechent et al. [16]	2015	Case Report	1	Tinea Incognito	Hyphae: refractive linear septate branching structures
Leclercq et al. [17]	2016	Case Series	2	Mucormycosis	Hyper-reflective elongated 20μm in diameter structures with perpendicular ramifications.
Uva et al. [18]	2018	Case Report	1	Tinea Nigra	Linear structures with high reflectivity and speckled appearance. Tortuous, irregular, and short hyphae structures, unlike thin and elongated dermatophytes.
Veasey et al. [19]	2019	Case Report	1	Tinea Capitis	Hyperreflectivity of structures with speckled appearance, as well as tortuous structures on the outside of hair
Pimenta et al. [20]	2020	Review	N/A	Tinea Candida Malassezia	Hyphae can be easily identified as bright, linear, branching and filamentous structures, whereas conidia appear as hyper-reflective small roundish bodies. Hyphae should be differentiated from the cell membranes of keratinocytes and from the normal structure of hair shafts. Candida is visualized with pseudo filaments and conidia in the skin, nails, and oral mucosa Malassezia appears as clusters of roundish bright structures with tortuous hyper- reflective structures corresponding to thick and short septa analogues of the typical spaghetti and meatballs description.
Cantelli et al. [1]	2021	Case Report	1	Tinea pedis	bright linear structures in the stratum corneum inflammatory infiltrate: high refractive polymorphonuclear clear clear cell populations, accompanied by light flashes in the epidermis
Lim et al. [21]	2021	Review	N/A	Onychomycosis	networks of bright filamentous septate hyphae
Potestio et al. [22]	2022	Case Report	1	Dermatophytid	several bright linear structures (mycelium) as well as inflammatory cells in the stratum corneum and in perivascular zones, as well as spongiosis.
Piccolo et al. [23]	2019	Letter	1	Majocchi granuloma	perifollicular inflammation, no hyphae
Ze-Hu Liu [24]	2019	Prospective study	10	Tinea favosa of the scrotum	septate branching hyphae
Cheng et al. [25]	2016	Letter/Case report	1	Tinea nigra	multiple, refractive, linear, filamentous structures at the stratum corneum
Cinotti et al. [26]	2014	Letter/Case Series	6	4: tinea capitis 2: tinea barbae	roundish homogeneous (5–10 μm in size) hyper-reflective structures corresponding to conidia around the proximal part of the hair shaft sparse elongated hyper-reflective structures outside the hair shafts, corresponding to hyphae
Rothmund et al. [7]	2012	Comparative study	60	Onychomycosis	white lengthy or thready structures with high reflection and typical shape or spore-like bright aggregates
Hongcharu et al. [27]	2000	Case report	1	Onychomycosis	branched hyphae
Markus et al. [28]	2001	Case report	1	Onychomycosis	several linear hyphae in intercellular spaces in the upper epidermis; highly refractile, linear structures that were brighter than the background keratinized cells

**Table 3 jof-09-00039-t003:** Clinical/Microbiological Diagnosis and Corresponding RCM Description.

Clinical/Microbiological Diagnosis	RCM Description
Tinea, unspecified [8,14,20]	bright [20], linear [20], branching [8,20], filamentous [20] fungal hyphae [8] in the epidermis [14], whereas conidia appear as hyper-reflective small roundish bodies [20]
Tinea manus [5]	bright linear branching structures not found in uninvolved skin (hyphae) with a variable number of bright linear structures distributed in the stratum corneum (mycelium) [5]
Tinea pedis [1,5,13]	bright [1,5,13], branching structures [1,5,13] not found in uninvolved skin (hyphae) with a variable number of bright linear structures distributed in the stratum corneum [1] (mycelium) [5]; inflammatory infiltrate: high refractive polymorphonuclear clear clear cell populations, accompanied by light flashes in the epidermis [1]
Tinea cruris [5]	bright linear branching structures not found in uninvolved skin (hyphae) with a variable number of bright linear structures distributed in the stratum corneum (mycelium) [5]
Tinea corporis [9,11]	bright [11], thin [11], branched [11], highly refractile [9], septate [9], linear structures [9,11]
Tinea incognito [10,16]	bright/refractive, linear, branching [10,16] thin structures at stratum corneum (hyphae) with aggregation of bright, small structures resembling inflammatory cells [10]
Onychomycosis [6,7,12,21,27,28]	several [28], bright [28], white [7], high reflective [7,12,21], lengthy [7,12], linear [28] or thready [7] or filamentous [6,21] structures in intercellular spaces in the upper epidermis [24] of the nail plate corresponding to septate hyphae [6,12,21]; spore-like, bright [7] roundish aggregates in the nail plate corresponding to arthroconidia [6]
Mucormycosis [17]	hyper-reflective elongated 20μm in diameter structures with perpendicular ramifications [17]
Tinea Nigra [18,25]	multiple [25], linear, filamentous structures with high reflectivity [18,25] and speckled appearance at the stratum corneum; tortuous, irregular, and short hyphae structures, unlike thin and elongated dermatophytes [18]
Tinea capitis [19,26]	roundish [26], homogeneous (5–10 μm in size) hyper-reflective [26] structures with speckled appearance [19] corresponding to conidia around the proximal part of the hair shaft [26]; sparse, elongated, hyper-reflective [26] tortuous structures outside of hair shafts, corresponding to hyphae [19]
Candida [20]	pseudo filaments and conidia in the skin, nails, and oral mucosa [20]
Malassezia [20]	clusters of roundish bright structures with tortuous hyper- reflective structures corresponding to thick and short septa analogues of the typical spaghetti and meatballs description [20]
Dermatophytid [22]	several bright linear structures (mycelium) as well as inflammatory cells in the stratum corneum and in perivascular zones, as well as spongiosis [22]
Majocchi Granuloma [23]	perifollicular inflammation, no hyphae [23]
Tinea favosa of the scrotum [15]	septate branching hyphae [15]
Tinea barbae [26]	roundish homogeneous (5–10 μm in size) hyper-reflective structures corresponding to conidia around the proximal part of the hair shaft [26]

## Data Availability

Not applicable.

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
