# Peer review of "Applications of Reflectance Confocal Microscopy in the Diagnosis of Fungal Infections: A Systematic Review"

_jof, 2022, doi:10.3390/jof9010039_

Round 1
Reviewer 1 Report
In this systematic review the authors describe application of reflectance confocal microscopy in the diagnostics of cutaneous and adnexal fungal infections. The topic is extremely relevant and presents non-invasive real-time diagnostic method for fungal infections so far diagnosed with microscopic preparation and culture with false negative and slow results, respectively.
Comments
1. The structure of the manuscript does not completely follow journal's demanded structure for systematic review (for example, description of cases is not usually part of the systematic reviews and the authors should think about removing them; however, Figure 2. and Figure 3. illustrate clearly described method and should be included in the manuscript)
2. The title of the manuscript is little misleading mentioning application in the both, diagnosis and therapeutic monitoring of fungal infections. However, therapeutic monitoring is not explicitly addressed in the manuscript and the authors should think about removing it from the title.
3. The same sentence is repeated twice in the Introduction section, line 35
Author Response
- The structure of the manuscript does not completely follow journal's demanded structure for systematic review (for example, description of cases is not usually part of the systematic reviews and the authors should think about removing them; however, Figure 2. and Figure 3. illustrate clearly described method and should be included in the manuscript. The description of cases is now removed. Figures 2 and 3 were also removed as they correspond to the case description.
-
The title of the manuscript is little misleading mentioning application in the both, diagnosis and therapeutic monitoring of fungal infections. However, therapeutic monitoring is not explicitly addressed in the manuscript and the authors should think about removing it from the title. The title of the manuscript was adjusted to only include diagnosis. We agree that more research should be done on the therapeutic monitoring aspect and this was not discussed extensively in our article.
3. The same sentence is repeated twice in the Introduction section, line 35. Noted and removed.
Reviewer 2 Report
Abstract: Does not provide an explicit main objective/ question that the review addresses, nor describes methods used to synthesize results. Bias and limitations are also not mentioned.
Introduction:
Lines 36-37 Statement repeated
Lines 40-42 Please rephrase the statement. It currently sounds as if fungal culture cannot easily distinguish from hair/wool/cotton fibers.
Lines 50-51: Statement is incorrectly quoted from the reference; culture is more specific.
Lines 54-70: Some sort of reference to the method should be provided. Reference 20-21 is given but it is out of sequence and not relevant.
There is no explicit statement or question that the review is addressing, just a review of cases identified by RCM. It does not state why the review is being done.
Methods:
It will be better if authors mention why meta-analysis was not performed, probably due to heterogeneity of studies.
There is no mention of any assessment for bias. At least some attempt should be made to go beyond the narrative and present cumulative data.
Results:
The table does not provide reference numbers to all the studies included in the review.
Results do not show the data collected on patient demographics, clinical diagnosis, diagnostic methods, histopathology, RCM methodology, RCM description, RCM advantages, RCM limitations, and fungal species.
Data on how the RCM findings correlated with clinical or microbiological diagnosis should be presented in some sort of cumulative form where available.
Discussion: Adequate
Conclusion: Study results cannot support these based on narrative alone.
Author Response
1. Abstract: Does not provide an explicit main objective/ question that the review addresses, nor describes methods used to synthesize results. Bias and limitations are also not mentioned.
Main objective now updated: We present an updated systematic review of the applications of RCM in diagnosing fungal infections in an effort to explore the utility of RCM as an adjunct clinical tool in detecting cutaneous and adnexal fungi.
Methods included: The following information on the application of RCM in the setting of fungal infections was extracted from each study, if reported: study type, year published, number of patients included, diagnosis/ diagnostic methods, and RCM description.
Limitation included: A limitation of this review is that 11 of 25 studies were case reports (n=1). Larger scale studies should be conducted to explore the utility of RCM in diagnosing fungal infections and to enrich the RCM descriptions of specific fungal conditions.
Introduction:
Lines 36-37 Statement repeated. Removed.
Lines 40-42 Please rephrase the statement. It currently sounds as if fungal culture cannot easily distinguish from hair/wool/cotton fibers. Rephrased: KOH scraping may also lead to false positive results as it may be difficult to distinguish fungal elements from contaminants (i.e., hair, cotton/wool fibers).
Lines 50-51: Statement is incorrectly quoted from the reference; culture is more specific. Corrected: The specificities of culture and KOH smear were 77.7% (72.2 to 82.5%) and 42.5% (36.6 to 48.6%), respectively [2].
Lines 54-70: Some sort of reference to the method should be provided. Reference 20-21 is given but it is out of sequence and not relevant. Reference 1 is now provided. The remaining content in that section s written by RCM pioneers Dr. Babar Rao and Dr. Navarrette- Dechent. But if not sufficient, we can certainly add an additional reference here. Thank you.
There is no explicit statement or question that the review is addressing, just a review of cases identified by RCM. It does not state why the review is being done.
Included now: We present a systematic review of the literature on the applications of reflectance confocal microscopy (RCM) in diagnosing fungal infections and we assess the utility of RCM in diagnosing cutaneous and adnexal fungal infections in clinical dermatology.
Methods:
It will be better if authors mention why meta-analysis was not performed, probably due to heterogeneity of studies. Mentioned that meta-analysis was not performed due to heterogeneity of results.
There is no mention of any assessment for bias. At least some attempt should be made to go beyond the narrative and present cumulative data.
Across all studies, clinical/microbiological diagnosis and their corresponding RCM descriptions were compiled by the two independent reviewers (BF and SK). Meta-analysis was not performed due to the heterogeneity of the studies.
Results: The table does not provide reference numbers to all the studies included in the review. Provided now.
Results do not show the data collected on patient demographics, clinical diagnosis, diagnostic methods, histopathology, RCM methodology, RCM description, RCM advantages, RCM limitations, and fungal species. Clarified in results which data was collected and reported: study type, year published, number of patients included, diagnosis/ diagnostic methods, and RCM description.
Data on how the RCM findings correlated with clinical or microbiological diagnosis should be presented in some sort of cumulative form where available. Presented in Table 2.
Discussion: Adequate. Updated with limitation.
Conclusion: Study results cannot support these based on narrative alone. Study results now adjusted to only support the data found in the narrative review.
Round 2
Reviewer 2 Report
The manuscript is now much improved. However, risk of bias assessment should be mentioned specifically in results and discussion as it is an essential component of systematic reviews. Simply stating that almost all studies were case reports does help but studies no. 7, 17, 19 can be assessed.
Here are a few online guides for assessing bias
Assessing risk of bias in included studies (cochrane.org)
9. Risk of Bias Assessment - Systematic Reviews - Research Guides at Utica College (libguides.com)
Author Response
Risk of bias assessment now using QUADAS-2 tool, included for studies 7, 17, 19:
Lines 121-129 and Lines 134-139
